# An anatomically informed computational fluid dynamics modeling approach for quantifying hemodynamics in the developing heart

**Kirsten Giesbrecht**[1], **Simone Rossi**[1], **Sophie Liu**[1☯], **Shourya Mukherjee**[1☯], **Michael Bressan**[2,3*], **Boyce E. Griffith**[1,3,4,5,6,7]

**1** Department of Mathematics, University North Carolina, Chapel Hill, North Carolina, United States of America, **2** Department of Cell Biology and Physiology, University of North Carolina at Chapel Hill, Chapel Hill, North Carolina, United States of America, **3** McAllister Heart Institute, University of North Carolina at Chapel Hill, Chapel Hill, North Carolina, United States of America, **4** Department of Applied Physical Sciences, University of North Carolina at Chapel Hill, Chapel Hill, North Carolina, United States of America, **5** Department of Biomedical Engineering, University of North Carolina at Chapel Hill, Chapel Hill, North Carolina, United States of America, **6** Carolina Center for Interdisciplinary Applied Mathematics, University of North Carolina at Chapel Hill, Chapel Hill, North Carolina, United States of America, **7** Computational Medicine Program, University of North Carolina at Chapel Hill School of Medicine, Chapel Hill, North Carolina, United States of America

☯ These authors contributed equally to this work.
* michael_bressan@med.unc.edu

## Abstract

Congenital heart defects occur in approximately 1% of newborns in the US annually. Currently, less than a third of congenital heart defects can be traced to a known genetic or environmental cause, suggesting that a large proportion of disease-causing mechanisms have yet to be fully characterized. Hemodynamic forces such as wall shear stress are critical for heart development and are known to induce changes in embryonic cardiac patterning leading to malformations. However, measuring these hemodynamic factors *in vivo* is infeasible due to physical limitations, such as the small size and constant motion of the embryonic heart. This serves as a significant barrier towards developing a mechanics-based understanding of the origins of congenital heart defects. An alternative approach is to recapitulate the hemodynamic environment by simulating blood flow and calculating the resulting hemodynamic forces through computational fluid dynamics modeling. Thus, we have developed a robust computational fluid dynamics modeling pipeline to quantify hemodynamics within cell-accurate anatomies of embryonic chick hearts. Here we describe the implementation of single plane illumination light sheet fluorescent microscopy to generate full three-dimensional reconstructions of the embryonic heart *in silico*, quantitative geometric morphometric methods for identifying anatomic variability across samples, and computational fluid dynamic approaches for calculating flow, pressure, and wall shear stress within complex tissue architectures. Together, these methods produce a fast, robust, and accessible system of analysis for generating

**Data availability statement:** This manuscript's minimal data set is hosted on the UNC Dataverse repository. Data can be retrieved via the following URL: https://doi.org/10.15139/S3/LLZOD1.

**Funding:** Research reported in this publication was supported in part by the North Carolina Biotech Center Institutional Support Grant 2016-IDG-1016 to the Microscopy Services Laboratory at UNC. IN addition, the UNC Hooker Imaging Core Facility and the Microscopy Services Laboratory are both supported in part by P30 CA016086 Cancer Center Core Support Grant to the UNC Lineberger Comprehensive Cancer Center. This work was also supported by the American Heart Association Predoctoral fellowship 899419 to KG, National Institutes of Health award R01HL146626 to MB, and National Institutes of Health awards R01HL157631 and U01HL143336 to BG, and National Science Foundation awards OAC 1652541 and OAC 1931516 to BG.

**Competing interests:** The authors have declared that no competing interests exist.

high-resolution, quantitative descriptions of anatomical variability and hemodynamic forces in the embryonic heart.

## Introduction

Congenital heart defects (CHDs) are the most common cause of birth defects in humans [1] and occur in approximately 1% of newborns in the United States [2]. Despite their prevalence, only 20–30% of congenital heart defects can be traced to genetic or environmental causes [3–5]. It is becoming increasingly clear that primary alterations in cardiac biomechanics and hemodynamics significantly impact proper morphogenesis of the heart and developing vasculature. Indeed, phenotypes mimicking human CHDs can be robustly reproduced by cardiac outflow tract banding or vasculature ligation in embryonic model organisms [6–9], suggesting perturbed hemodynamics contribute to abnormal cardiovascular development and function. Particularly, studies have shown that wall shear stress (WSS) regulates several aspects of cardiogenesis through mechanosensitive pathways [10–14].

The relevance of factors such as WSS is becoming increasingly emphasized as critical for heart development. However, obtaining accurate measurements or experimentally determining regional variations of shear stress are both prohibitively difficult due to the minuscule size, delicate nature, and continuous pumping of the embryonic heart. Thus, we currently lack experimental tools that can quantitatively describe spatial and temporal intracardiac hemodynamics which precludes direct modeling of how perturbations to these features would contribute to altered development. Computer modeling approaches for assessing fluid forces, such as computational fluid dynamics (CFD) simulations, offer tractable alternatives to direct experimental measurements in animal models of four-chambered hearts [15–20].

Ultimately, this fast pipeline for comparing morphological features and variation to hemodynamic patterns in the embryonic chick heart model can be extended to other model systems in order to predict how differences in individual morphology and hemodynamics can impact heart development. Additionally, it is fully expected that computational modeling approaches can be used in the future to investigate hemodynamics in human heart development, as fetal cardiac imaging improves [21]. Given the similarities in size, structure, and function between human and chick heart development, the chick serves as an ideal model system to run proof-of-principle analysis for such techniques.

To date, however, a major barrier to the broad application of such CFD models to cardiovascular developmental research has been the lack of protocols for generating cell-accurate descriptions of the embryonic cardiovascular anatomy through which to run simulations. Herein, we use simulations of blood flow and the resulting shear stress throughout the heart and surrounding vasculature to capture spatial and temporal patterns of shear stress [22] informed by cell-accurate imaging of a cohort of embryonic geometries.

Evaluating large-scale three-dimensional (3D) imaging data sets across large collections of samples poses new challenges compared to traditional histological

analysis and requires harnessing tools than have not been traditionally used to examine phenotypic differences in developmental biology. There is no consistent method in developmental biology to analyze volumetric data quantitatively. However, using morphometrics to evaluate 3D spatial data is routinely used in other fields, namely paleontology, evolutionary biology, and anthropology [23]. Morphometrics is a statistics-based system of evaluating a specimen's shape and size [23] using data of structural lengths and angles for classification purposes [24]. By the 1980s, geometric morphometrics emerged from traditional morphometrics methods to quantify more complicated anatomical features, such as vertebrate skulls, using spatial landmarking approaches [23,24]. Geometric morphometrics has been used in a handful of developmental biology studies, including phenotypic response to gene expression manipulations [25], but it has not yet been widely adopted by the field. However, given the rapid evolution of high-resolution multidimensional imaging approaches, utilizing geometric morphometric methods is of high potential value for conducting unbiased anatomical phenotyping. Furthermore, the quantitative anatomical comparisons generated by geometric morphometric methods are ideal for analyzing how hemodynamic features vary given differences in cardiac or vascular geometry.

Thus, the aim of this study was to develop a robust and accessible biocomputational pipeline to capture and compare variations in geometric features and resulting shear stress distributions within intracardiac regions prone to congenital heart defects[26]. Specifically, we designed this pipeline with the intent of constructing high-resolution, cell-accurate descriptions of local anatomy in which analysis of flow dynamics could be rapidly performed within cohorts of healthy or perturbed cardiac anatomies (Fig 1). To accomplish this, we have combined techniques for microinjection-based intravital labeling of the embryonic vasculature with light sheet microscopy. We then extracted the vascular geometries of these embryos to simulate blood flow through them. To our knowledge, this is the first study utilizing light sheet fluorescence microscopy (LSFM) to reconstruct subject-specific embryonic chick heart anatomies for computational simulation. This is significant because the relative ease of use and availability of light sheet imaging systems promise to enable high-throughput studies of anatomical variation in broad sample groups, largely eliminating the need to perform simulations with idealized geometries. Herein, we use the full incompressible Navier-Stokes equations to simulate blood flow. These equations are numerically solved using the finite element method, which is particularly well-suited for models involving complex geometries [27]. Additionally, we use morphometrics methods to unbiasedly examine geometric variation across the cohort of three-dimensional geometries. Together, the techniques described below provide a powerful new system to evaluate previously challenging hemodynamic features of the embryonic heart.

## Results

### Model system selection

Chick embryos are attractive experimental models for studying normal and pathological blood flow in cardiogenesis [8,14,16,28–31] due to their similar order of developmental events [28,32] and comparable size to the human fetal heart, as well as their ease of manipulation and visualization [33,34]. Unlike several other model organisms such as zebrafish, which have a two-chambered heart, chicken hearts have four chambers, as do human hearts. Due to these similarities, especially when considering hemodynamic environments, the chick embryo was selected as the model organism for this study.

Intracardiac and cardiovascular hemodynamics in embryonic chick hearts have been frequently modeled in many CFD studies [15–17,19,20,35,36]; however, most of these intracardiac hemodynamics studies employ idealized geometries or combine several individual anatomies to form a single representative geometry of a developing chick heart [19,37]. Comparatively few studies reconstruct individualized anatomies from distinct embryonic chick hearts, but these cohort-based studies have reported that areas displaying anatomical variability across embryos do impact local shear stress. Unfortunately, this level of detail is often lost in subject-averaged models [15], and different methods of constructing local tissue geometries have yielded wide-ranging data on a host of basic morphological features such as average aortic arch (AA) midpoint diameter. These studies also report a broad range of shear stress patterns, and we have summarized a

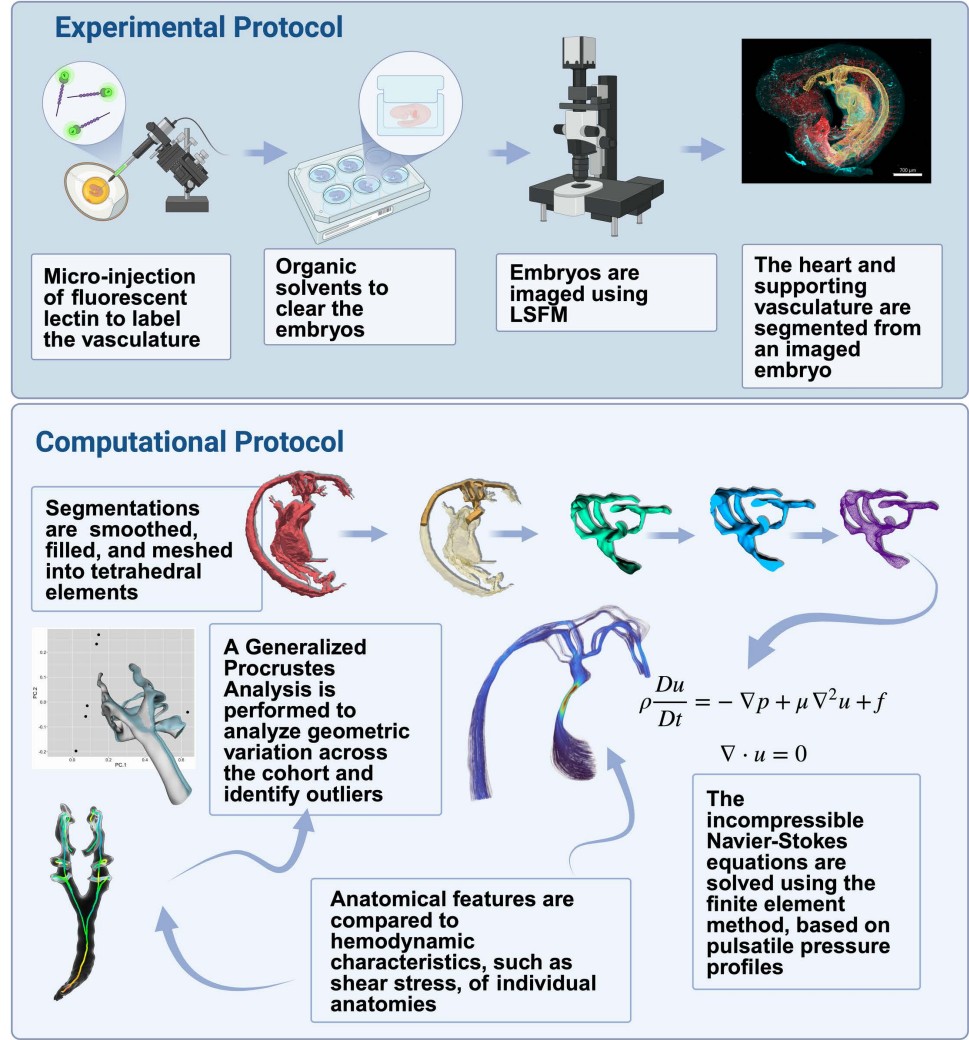

**Fig 1. Overview of combined experimental and computational pipeline to compare hemodynamic environments and shape variation across a cohort of embryonic chick hearts.**

non-exhaustive list of examples of reported geometric values and WSS ranges that different imaging techniques of embryonic chick AA CFD models have yielded (Table 1).

We chose to simulate blood flow through the aortic arches and the outflow tract, as these are traditionally difficult to image regions and are prone to congenital heart defects [16,32,39,40]. Most congenital heart defects are concentrated in the outflow tract [16], a transitory region that connects the early heart to the aortic arches (AAs) [26] and eventually gives rise to the pulmonary trunk and aorta. Congenital defects occurring in the outflow tract have a high morbidity and mortality rate [26]. The AAs undergo significant remodeling during embryonic growth and contribute to the aorta and pulmonary artery [35]. Generally, three or four pairs of AAs are present at a time, and 3 pairs are present upon completed cardiac development [41]. Because they undergo extensive remodeling during development, the arches are prone to severe congenital heart defects [40]. The outflow tract and AAs are sensitive to hemodynamic perturbations [32] and are the sites for half of the congenital heart defects in newborns [32,39].

**Table 1. Reported HH18 Embryonic Chick Morphological and Hemodynamic Properties.**

| Imaging Method | Anatomy Reconstruction Approach | Type of AA Measurement | Measure-ment (mm) | WSS Ranges (dynes/cm²) |
|---|---|---|---|---|
| Nano-CT [15] | Collection of individual anatomies | Maximum cross-sectional area | 0.0521 | 80-160 |
| Micro-CT scanned resin casts [37] | Combined multiple scans to create representative baseline geometry, with arteries added manually | Midpoint diameter | 0.075-0.129 | 0-687 |
| Immuno-fluorescence [38] | Measured lumen area of each AA, with cross sections considered circular | Average diameter | 0.28-0.3 | Mean: 20 |

To inform the fluid domain of the CFD model we sought to build, we used the chick embryo to create 'cell accurate' geometries of the embryonic heart and the proximal vasculature. In addition, we wanted to utilize a technique for generating this geometry that was adaptable to various experimental techniques (such as outflow banding or vascular ligation) and was compatible with live-imaging approaches. Due to these criteria, we selected live cell labeling using *Lens culinaris* agglutinin (LCA). LCA is a member of the lectin family of carbohydrate-binding factors which labels glycosylated moieties present on the surface of endothelial cells. LCA has a high affinity for chick vascular endothelial cells at early stages of development, when sensitivity to blood flow perturbations on cardiac morphogenesis is believed to be high [42,43], and can also be used at later stages of chick development [44]. Therefore, this labeling protocol can easily be adapted to any age embryo. Furthermore, unlike wheat germ agglutinin and other lectins, LCA has the ability to bind to finer vessels, appropriate for this early embryonic stage and the resolution we aimed to capture in imaging [44]. To increase utility for deep tissue imaging, we utilized LCA conjugated to the far-red fluorescent fluorophore DyLight 649. For the applications noted in this report, we focused on generating vascular geometries from Hamilton Hamburger (HH) stage 16–17 embryos (approximately embryonic day 2.5). Herein, LCA (1:10 diluted in Hanks Buffered Salt Solution+/+) was back-loaded into a pulled glass microcapillary needle (1.0 OD/0.7, World Precision Instruments) and 2–5 µl were pressure injected into circulation via the extraembryonic vitelline vein. A protocol of pulsed injections (200 kPa) using a FemtoJet Microinjector (Ependorf) was used as the delivery system. Using this approach, vascular and heart labeling can be immediately confirmed using a standard fluorescent dissecting microscope.

## Clearing whole embryos housed in agarose containers for imaging

Based on the overall tissue dimensions that we sought to digitally reconstruct for CFD simulations, we selected LSFM as a capture method due to its speed and high resolution [45,46]. LSFM also minimizes photobleaching and light scattering compared to alternative imaging techniques such as confocal microscopy [45], and is better at imaging soft tissue when compared to micro-CT [47]. While light sheet microscopy has been proposed for use and implemented in CFD studies in zebrafish hearts [48,49], currently it is not routinely implemented for this purpose in four-chambered hearts.

To prepare samples, we utilized a modified clearing technique to preserve LCA staining. We selected the iDISCO+ protocol as the clearing method versus other options due to its low cost, speed, and compatibility with LSFM [50]. We cleared whole embryos using the iDISCO+ organic solvent-based clearing protocol with the Alternative Pretreatment [51] to clear whole embryos.

## Imaging whole embryos using light sheet fluorescence microscopy

After being cleared, embryos were imaged on a Lavision BioTec Ultramicroscope II light-sheet system. The pixel size was between 1.2–1.95 µm, with each Z slice spaced 5 µm apart and a horizontal focus of 5 µm. To excite the stained lumen and autofluorescence, both 647 and 488 nm laser excitation was used to image two channels, respectively, for each embryo. The Imaris (Bitplane, Oxford Instruments) File Converter software was used to convert light sheet fluorescence microscopy-generated TIFF files to Imaris files.

## Image segmentation

Following image acquisition, cardiovascular anatomy needed conversion to a format usable for CFD modeling. For these purposes, we used Imaris software (Bitplane, Oxford Instruments) [52] to visualize and manually segment the embryonic heart and vasculature geometries. The LCA 649 fluorescence channel (Fig 2A) combined with the autofluorescence channel (Fig 2B) were used to define vascular and cardiac lumen structure (Fig 2C). As shown in Fig 2D-F, this resolves fine details related to vascular bifurcations, microvessel anatomy, and cardiac chamber arrangement. As the cardiac outflow tract and AAs have previously been examined using CFD models, we focused on these areas of the heart to confirm the viability of our approach. Typically, imaging the AAs is difficult due to their complicated geometry, small size, and dorsal/anterior position relative to the looping heart, so alternative methods such as ink injections have been used to map the arches [37]. However, we observed high fidelity and reproducibility across samples when imaging these regions with light sheet fluorescence microscopy. Additionally, LCA 649 labeling allowed us to generate high-resolution segmentations of both the outflow tract and AAs despite their complex configurations and to clearly distinguish the lumen of the outflow tract and developing chambers from soft tissue within the heart.

To translate raw imaging data into three-dimensional geometries appropriate for CFD modeling, we generated a processing pipeline to build structural scaffolds matching the vascular anatomies obtained by light sheet imaging (S1 Fig A). As an example of this process, we extracted aortic arch geometric patterns and prepared a cohort of these samples for CFD simulations. Initially, we cropped to a region of interest containing the aortic arches, outflow tract, and dorsal aorta. Then, each "hollow" vasculature structure was post-processed in the software MeshMixer (Autodesk) [53] to resolve rough boundaries remaining from manual segmentation. Next, the hollow segmentations were translated into solid structures in Blender (The Blender Foundation) [54] using the Quad Remesher plugin (Exoside) [55] which retopologizes the geometry to make it suitable for meshing. Following this step, the AAs, outflow tract, and dorsal aorta were meshed and refined using Cubit (Coreform) [56] into at least 100,000 second-order tetrahedral elements for each anatomy to be used for CFD simulations. The grid spacing was approximately 0.009 mm. We conducted a mesh convergence study (S3Fig A-B). Upon completion of segmentation discretization, we analyzed the geometric features of each embryo in the sample to evaluate our geometries in relation to previous reports.

## Geometric morphometrics analysis

In geometric morphometrics methods (GMM), landmarks or pseudolandmarks (pseudoLMs) defined using Cartesian coordinates are assigned as homologous reference points across a group of specimens and used to compare geometric similarities and differences within a sample [57]. There are various GMM approaches, with the most common being the Procrustes method or generalized Procrustes analysis (GPA) in biological fields [23,24,58]. The GPA involves standardizing landmark position, scale, and orientation for each set of landmark coordinates for every specimen by translating, scaling, and rotating the landmarks to a common centroid using a least squares approach between each specimen and a mean shape [23,25]. Dimensionality reduction techniques, such as Principal Component Analysis (PCA), are often used in morphometrics analyses to visualize results of highly complex multidimensional datasets produced from the GPA. Visualization methods such as mean shape plotting and PC scatter plots can be used to interpret the biological results of the data [23]. Here, we apply the GPA to evaluate similarities and differences that emerge across a cohort of embryonic chicks to understand what natural variation is expected and to illustrate GPA sensitivity to geometric differences.

We performed a GPA on the cohort of embryos which included two specimens lacking at least one AA or cranial arch due to tissue tearing during processing and imaging. This tested if the GPA could distinguish specimens with missing AAs from structurally normal embryos. Using the software 3D Slicer (https://www.slicer.org/) [59] and the 3D Slicer extension SlicerMorph [60], we placed 275 pseudolandmarks [61] defined using curves along geometric features such as the AAs and contours of the outflow tract along a randomly chosen template specimen from embryos containing six arches (Fig 3A). After the pseudoLM set had been defined on the template specimen chosen from a control case, the set was

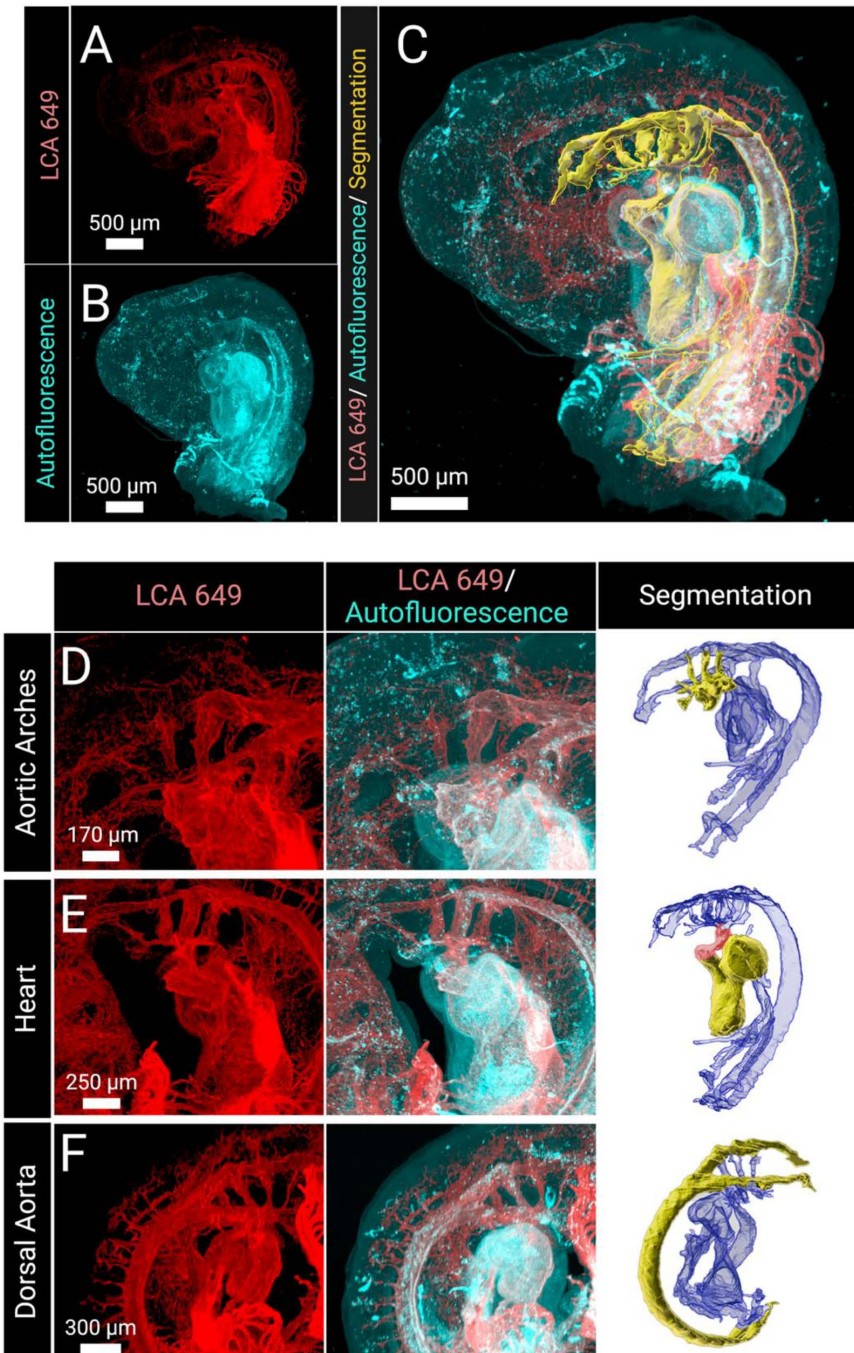

**Fig 2. Light sheet fluorescence microscopy optically sections whole embryos for generating anatomy reconstructions.** (A) The LCA 649 channel distinguished endothelial cells lining the lumen and was primarily used to inform segmentations. (B) The autofluorescence channel distinguished soft tissue and was used to verify segmentations. (C) Together these channels were used to construct segmentations of the developing heart, AAs, and dorsal aorta. (D-F) The imaging captured anatomical features that are often difficult to obtain, due to their size and position. In the segmentation panel, the segmented portions of interest are highlighted as solid yellow within the entire segmented anatomy (blue). (D) AAs, which are notoriously difficult to image due to their small size and position, were segmented using the LCA 649 channel. (E) The segmented heart and outflow tract (red). The LCA channel distinguished the lumen from other soft tissue, such as the cardiac jelly, which can be seen in the autofluorescence channel. (F) The bifurcating dorsal aorta segmented using the LCA 649 channel.

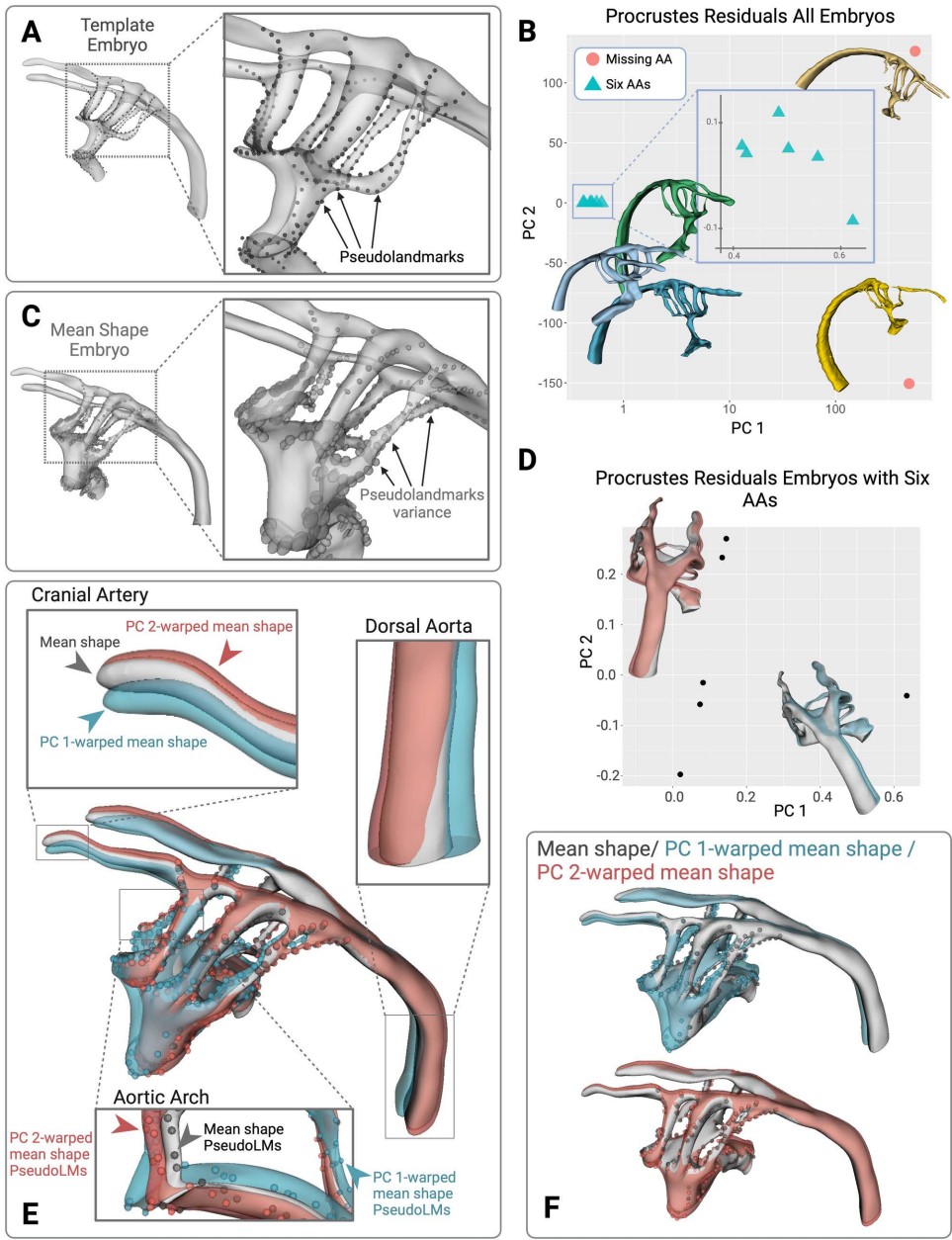

**Fig 3. Geometric Morphometric Methods can be applied to segmented embryo cohorts and used to identify outliers and variation across individuals.** (A) A template embryo was selected and 275 pseudolandmarks were placed along the contours of geometric features, such as the AAs, for the analysis. (B) Originally, two embryos that were missing multiple AAs were included in the GPA. The PCA scatter plot of the Procrustes residuals separates these two embryos from the rest of the cohort, with an inlay plot showing the detail. X-axis is shown in log scale. (C) The variance of each pseudolandmark is plotted by representing each as an ellipsoid. The radius length of each ellipsoid corresponds to the variance in that dimension of the pseudoLM. (D)The GPA was repeated on only embryos containing all six AAs. The PCA plot displays how the embryos varied in PC 1 and PC 2. The mean shape (grey), and the mean shape warped by PC 1 (blue) and PC 2 (red) is displayed. (E) Detailed views of the different anatomical regions of the mean shape, pseudolandmarks and warped mean shapes, and warped pseudoLMs. (F) View comparing the mean shape and PC 1-warped mean shape and PC 2-warped mean shape, pseudoLMs, and warped pseudoLMs.

transferred from the template model to all the specimens using the SlicerMorph module ALPACA (Automated Landmarking through Point Cloud Alignment and Correspondence Analysis) [62].

After generating pseudolandmarks for all specimens, we performed the GPA and preserved Boas coordinates to maintain size variation using the corresponding module in the SlicerMorph extension. We observed that the embryos containing all six arches clustered together and were separated from the embryos lacking at least one arch in the PCA scatter plot of the Procrustes residuals (Fig 3B), demonstrating the GPA is well-suited to distinguish gross anatomical differences among specimens.

Next, we removed the outliers containing the missing AAs to repeat the GPA with the six phenotypically normal embryos, then plotted the PCA scatter plot of the Procrustes residuals. We visualized these results by plotting the mean shape resulting from the GPA and warped the mean shape by PC 1 and PC 2, separately. The mean shape and warped mean shape are displayed on the PCA plot, indicating how the specimens vary with respect to PC 1 and PC 2 (Fig 3D). Fig 3E shows the relative variance of each pseudoLM in each dimension as indicated by the radius length of the corresponding dimension of ellipses. The results indicate relative variance was not uniform across the pseudolandmarks, as they varied greatly along the outflow tract and left AA II. In Fig 3F the mean shape (grey), warped by PC 1 (blue), indicates that PC 1 corresponds to a shift in the outflow tract to the left and a twisting about the AAs which resulted in the dorsal aorta shifting to the right. Additionally, PC 2 (red), indicates a downward shift and shortening of the outflow tract, resulting in the dorsal aorta rotating to the right. One specimen was more positively associated with PC 1 compared to the other specimens considered. This analysis demonstrates that GPA represents a completely unbiased method of distinguishing anatomical variation across large numbers of complex three-dimensional imaging datasets for cohort-based studies. The biological interpretation of the PCA plot of the Procrustes residuals indicates that specimens are likely to vary by outflow tract shape, AA II, and the angle of the dorsal aorta to the AAs. The GPA also highlights that shape variance is not uniform across all geometric features in the sample group.

## Basic geometric and centerline analysis

In addition to the GPA, we also analyzed basic geometric features of embryos. We observed that the AA cross-sections are elliptical, generally with the longer axis oriented laterally (left to right) (S1 Fig B) and that the central AA III pair typically had the largest diameter (S1 Fig C) which is consistent with reports from previous studies that analyze individual embryonic heart anatomies across a cohort from nano-CT imaging data [15]. To measure quantitative geometric features, we extracted centerlines through the dorsal aorta and aortic arches of each embryo (S1 Fig A-B) using the vascular modeling toolkit (VMTK) extension [63] in the open-source image computing platform, 3D Slicer (https://www.slicer.org/) [59]. Comparing AA geometric features revealed that AAs primarily varied geometrically by pair but not by side, and several other trends were noted (S1 Fig D-G). The centrally located arch pair, AA III, was significantly shorter than either of its neighboring arches (S1 Fig D). AA IV exhibited the highest degree of stenosis, as measured by the minimum reported diameter across each arch (S1 Fig E). The average diameter of AA III was significantly larger than the average diameter of AA IV (S1 Fig F). However, the volume of each aortic arch did not vary significantly by arch pairs or sides at this stage (S1 Fig G). This pattern of AA IV arch pair having the smallest average diameter and AA III having the largest average diameter is consistent with previous reports among the diameters of these three arch pairs measured at similar stages [37].

## Embryonic heart anatomical variability

Following the geometric analysis of each embryonic heart anatomy, we noticed variability in vessel configurations across the embryos. All embryos examined had bilaterally paired aortic arches, but the degree of stenosis in the outflow tracts and AAs varied across specimens. Additionally, we observed a geometry (hereafter referred to as case 1) where the AA II pair had a larger diameter than the AA III. We also observed a geometry (case 2) where there was an outlet in each cranial artery. Section 'CFD simulations' describes how blood flow varied in these special cases.

## CFD modeling parameters

To implement a flexible CFD modeling pipeline that can be used to robustly simulate anatomy specific hemodynamic characteristics, we defined a basic set of blood flow traits and pressure profiles for use in cardiac simulation. Initially, we focused on simulating blood flow about the AA anatomies as these vessels showed a high degree of variability across our sample cohort and are particularly susceptible to congenital malformation. Herein, we modeled blood as an incompressible Newtonian fluid with a viscosity, $\mu = 3.71 \cdot 10^{-3}$ Pa·s and density, $\rho = 1060$ kg·mm$^{-3}$ as reported by a previous embryonic blood rheology study [64]. We used the full incompressible Navier-Stokes equations, including the nonlinear convective term. The momentum equation described blood flow (equation 1) and the incompressibility constraint (equation 2), in which $p$ and $\boldsymbol{u}$ are the fluid pressure and velocity fields, respectively:

$$\rho \frac{Du}{Dt}(\boldsymbol{x}, t) + \nabla p(\boldsymbol{x}, t) - \mu \nabla^2 \boldsymbol{u}(\boldsymbol{x}, t) = \boldsymbol{f}(\boldsymbol{x}, t),$$

(1)

$$\nabla \cdot \boldsymbol{u}(\boldsymbol{x}, t) = 0.$$

(2)

We solved the system of equations using the finite element method as it facilitates models involving complex geometries [27]. The 3D CFD simulations to model blood flow were performed using an in-house C++ code based on the libMesh [65] finite element library and using linear solvers implemented in PETSc [66]. This technique required a mechanism to stabilize pressure [67]. To this end, we used LBB-stable P2-P1 Taylor-Hood velocity-pressure pairs [67,68]. For our simulations, we set the grid spacing to be approximately 0.009 mm, with each anatomy containing at least 100,000 second-order tetrahedral elements. Previous studies have calculated Reynolds numbers to be between 0.1 to 9.0 for the stages of development we have analyzed [15,69–71] (1,2,3,4). This would indicate that intracardiac blood flow is highly laminar during early cardiac morphogenesis [19,72,73].

No slip boundary conditions were imposed on intracardiac and vessel walls, which are assumed to be rigid and impermeable. Sinusoidal pulsatile pressure profiles, determined by digitizing previously reported pressure profiles in the ventricle and dorsal aorta [74], were imposed at the inflow and outflow boundaries, respectively. Simulations and resulting hemodynamics were visualized in ParaView (Kitware) [75].

## CFD simulations

Using the above parameters, we determined how geometric features that varied across our sample cohort impacted hemodynamic patterns within the AAs. WSS was examined both at peak flow (Fig 4A) and during the accelerating phase of blood movement (Fig 4B). In general, elevated WSS was observed in stenosed regions of the anatomies. Consistent with our rationale for using a cohort-based approach, we noted that the regions of maximal WSS varied with local AA and outflow tract geometry (Fig 4C-E). Conversely, if the outflow tract was heavily stenosed, elevated WSS occurred in the outflow tract (Fig 4D). If the outflow tract was moderately stenosed, elevated WSS was observed throughout locally stenosed regions, including the outflow tract and the AAs (Fig 4E). In addition, spatial WSS patterning across embryos was similar during accelerating flow and at peak flow time points (Fig 4A-E). These similar WSS patterns indicate the flow is not heavily driven by convective effects and pulsatile hemodynamics of the cardiac cycle do not shift the positions of highest WSS. This is consistent with laminar flow and previous reports of laminar flow in CFD models of this region [19]. As expected, peak WSS occurs during peak flow and the pressure difference between the dorsal aorta and outflow tract is highest at this time (Fig 4F). The ranges of WSS we observed are similar and overlap with previous WSS ranges reported [15,37].

Unlike an experimental approach, our CFD modeling can sample hemodynamic features at any position within the acquired geometry. This allowed us to determine that the central pair of AAs (AA III) displayed the highest average WSS across our samples (Fig 4G). AA III was the shortest pair present in our embryos but was not the most stenosed arch pair

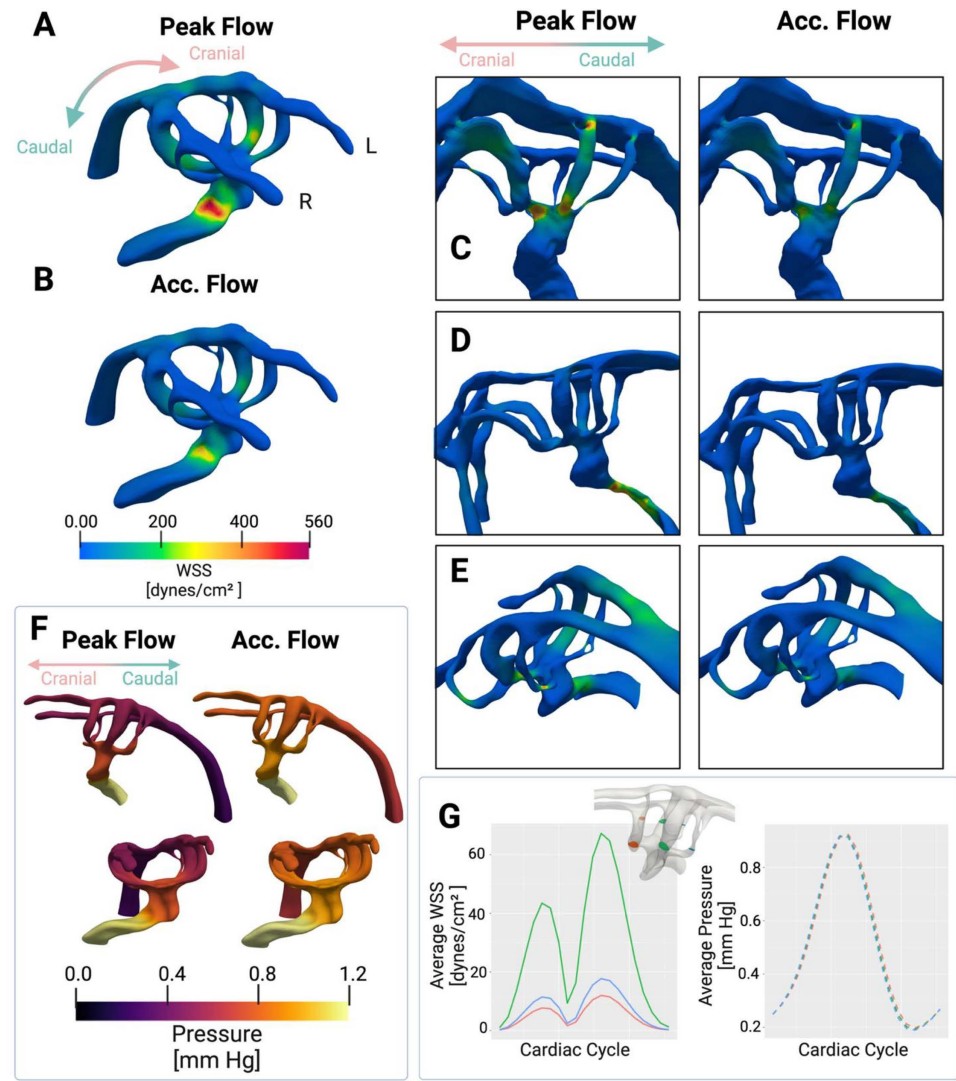

**Fig 4. Wall shear stress is elevated in locally stenosed regions such as the AAs.** (A) (A) WSS was observed during peak flow and (B) accelerating flow. As expected, peak WSS occurred during peak flow. Peak WSS locations varied by individual anatomical features within embryos. (C) In the case 1 embryo with less stenosis in the outflow tract, peak WSS occurred in the AAs. (C) Conversely, in the case 2 embryo with a highly stenosed outflow tract, peak WSS occurred in the outflow tract. (E) An embryo with a moderately stenosed outflow tract experienced elevated WSS throughout locally stenosed regions including the outflow tract and AAs. WSS spatial patterning is similar during peak flow compared to accelerating flow. Regions that experience the highest WSS during peak flow experience high WSS during accelerating flow, indicating that this flow regime is not heavily driven by convective effects. (F) The pressure difference between the outflow tract and dorsal aorta was larger during peak flow compared to accelerating flow, as expected. (G) Average WSS was highest in the AA III pair, which is the centrally located of the three pairs present during this stage of development, despite all pairs of the AAs experiencing consistent pressure and the higher degree of stenosis in flanking arches of the AA III pair.

(Fig 4D-F). Of note, each AA pair experienced similar pressure during the cardiac cycle, despite the central pair undergoing higher WSS (Fig 4F, G).

A potential explanation for the higher WSS in AA III compared to the flanking arches would be that flow distribution in the arches is not homogeneous. To further examine this, we extracted blood velocities as flow streamlines calculated through the AA anatomies and found that blood flow velocity streamlines were consistent with WSS patterns. Specifically, blood flow velocity was elevated in locally stenosed regions of the embryonic heart where peak WSS was found, such as

the outflow tract and AAs (Fig 5A). Elevated velocity was observed in AA III, consistent with the elevated WSS patterns (Fig 4A, 5A). We observed that blood flow velocity patterns, including peak velocity locations, were consistent with peak WSS patterns across embryos (Fig 5B-D). The spatial patterning of velocity streamlines was also maintained across acceleration and peak flow (Fig 5). As expected, this is consistent with the spatial WSS patterns maintained during the two different stages of flow. Since the location of peak velocity streamlines is maintained across different flow phases, this further supports the finding that there is little convective mixing and that flow is laminar.

To examine how blood traveled through the AAs, the streamlines were traced through each A. We placed a point cloud enclosing each AA cross section to seed each streamline. Next, we traced the streamlines of each AA from the outflow tract to the dorsal aorta. The streamline paths were compared from the outflow tract through the dorsal aorta. We observed the streamlines which flowed through the AA IV pair initially flanked each other in the outflow tract but separated to opposite sides in the cross section of the dorsal aorta after traveling through the arches. Streamlines that contributed to the AA II pair also flanked each other in the outflow tract but did not separate in the dorsal aorta (Fig 6A-D);

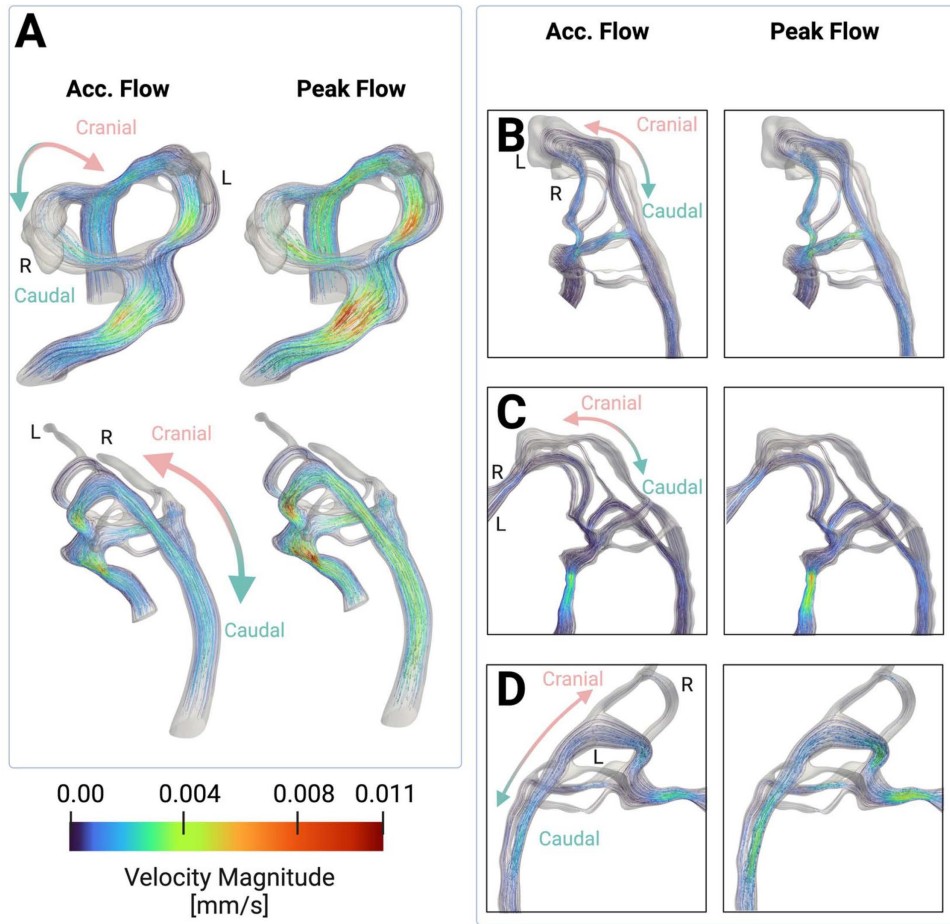

**Fig 5. Blood flow velocity streamlines are consistent with WSS patterns.** (A) Blood flow velocity streamlines are observed during peak and accelerating flow, exhibiting similar temporal and spatial patterns compared to WSS, consistent with the finding that there is low convective flow. As expected, blood flow velocity streamlines indicate elevated flow in the same regions as elevated WSS across variations in embryonic anatomies. (B) An embryo with little stenosis in the outflow tract experiences elevated velocity primarily in the AAs, consistent with WSS patterns. (C) An embryo with highly stenosed regions in the outflow tract experiences elevated flow velocity in the outflow tract, consistent with the elevated WSS. (D) An embryo with moderate outflow tract stenosis experiences elevated WSS throughout locally stenosed regions in the outflow tract and in the AAs.

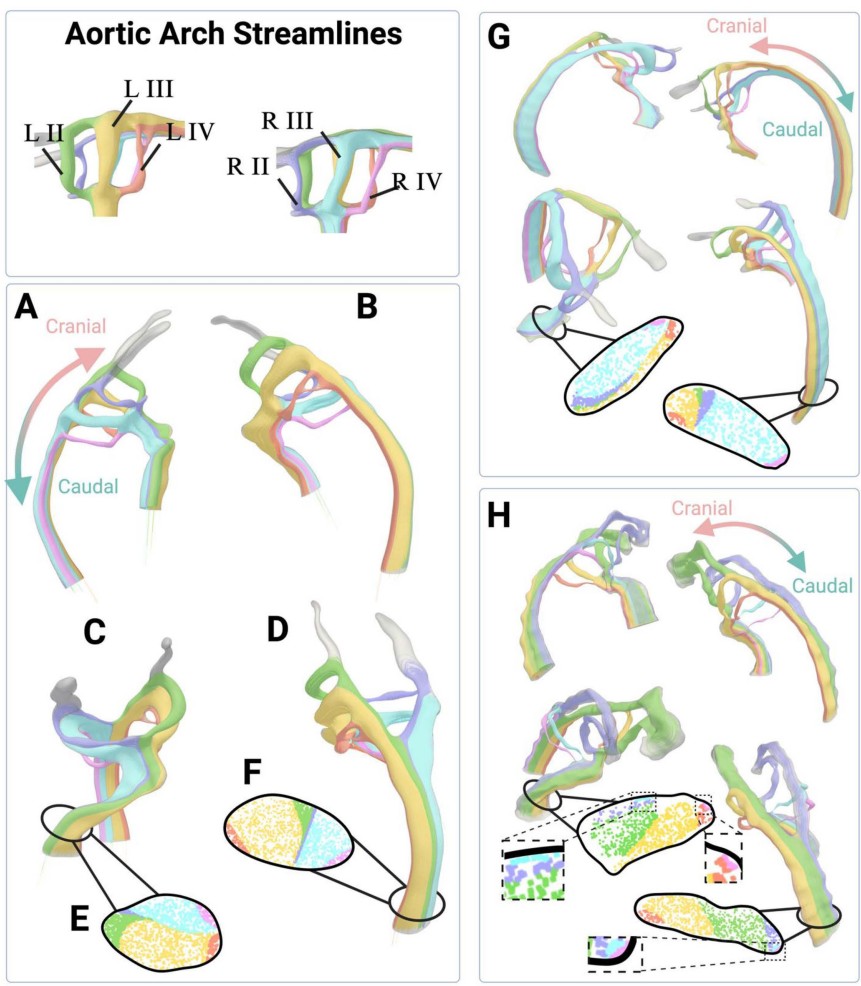

**Fig 6. Streamline tracing confirms that there is low convective flow.** Streamlines identified through each aortic arch. Paths were traced from the outflow tract to the dorsal aorta to understand streamline spatial configuration in relation to before and after arch flow. Streamlines through the left AAs (A) and right AAs (B) of an embryo show that there is little mixing of flow streams, even after exiting the AAs. [C-D] Two views of the dorsal aorta demonstrate that the streamlines through AA IV pair are split by the other arch pair streamlines while traveling down the dorsal aorta. (E) We examined streamlines in the cross sections in the outflow tract, upstream of the AAs, (F) and in the cross section of the dorsal aorta, downstream of the AAs. The order of the streamlines in the outflow tract was maintained in the dorsal aorta, and confirms that the AA IV pair separates after flowing though the aortic arches. There is minimal mixing of streamlines, consistent with the finding that there is little convective influence in this flow regime. (G) A second embryo illustrates the spatial order of streamlines is largely conserved across anatomical variations. (H) Examined streamlines through an embryo where AA III is smaller than AA II, showing that the distribution but not order of streamlines is altered. Detailed views of the cross sections are shown to highlight all the AA streamlines.

instead, they traveled towards the center of the dorsal aorta cross section. Examining the cross section in the outflow tract (Fig 6E) and of the dorsal aorta (Fig 6F) confirms these patterns. The cross sections of the streamlines demonstrate that flanking AA streamlines in the outflow tract generally maintain their flanked configuration in the dorsal aorta, with the exception of the AA IV paired streamlines which separate. For instance, the R AA II streamlines are flanked by the L AA II and R AA III streamlines in the outflow tract (Fig 6E) and reunite with these two sets of streamlines in the dorsal aorta (Fig 6F). This maintenance of neighbors of streamline sets from the outflow tract to the dorsal aorta was consistent across embryos (Fig 6G) and is also consistent with laminar flow.

This spatial order of streamlines from the outflow tract to across the dorsal aorta was maintained even in embryos with unusual variation in the AAs. For instance, in case 1, the dorsal aorta cross section contained the same spatial order of streamlines, despite the difference in flow distribution across the arches (Fig 6H). In case 2, we observed that the streamlines from the AA II pair contributed to the cranial artery outlets instead of the dorsal aorta outlet. The streamlines from the remaining arches maintained the same spatial ordering in the dorsal aorta cross section (S2 Fig). These data demonstrate there is a predictable pattern that streamlines follow before and after entering the aortic arches that is conserved across anatomical variations.

## Discussion

In this report, we have described a fast, accessible, and high-throughput pipeline for reconstructing cell-accurate surfaces of embryonic chick heart lumens that can be used in CFD models of blood flow and geometric analysis. Additionally, we demonstrate that unbiased GMMs can identify embryos that greatly vary phenotypically, as shown in the GPA of the embryo cohort including embryos that lacked typical anatomical features for the developmental stage. As imaging techniques continue to evolve, highly detailed, multi-dimensional datasets are becoming the norm. However, standardized techniques for performing unbiased geometric analysis for conducting morphometric comparisons across sample groups have largely not been identified or implemented. This is particularly problematic in the developing heart, which is a complex, three-dimensional structure that dynamically changes its size, shape, and alignment across development. Thus, in this report we demonstrate how GMM may address a large gap in the field. Furthermore, we pair the geometric analysis with CFD models to relate specimen-specific geometry to hemodynamic patterns. This pairing of GMM with CFD models could also be applied to predicting critical morphological differences that lead to altered blood flow dynamics and impact human heart development. As imaging for fetal hearts rapidly improves, having a detailed understanding and robust method for analyzing this relationship between morphology and hemodynamics already established would be an extremely useful tool.

We labeled the lumen of the embryos using LCA, which has a higher affinity for embryonic chick endothelial cells throughout development compared to other lectins [44]. Consequently, this pipeline can easily be extended to imaging and assessing embryonic chick geometries and hemodynamics throughout various stages of development. We employed light sheet fluorescence microscopy, requiring only minutes of acquisition time, giving a fast and accessible imaging method to capture the difficult-to-reach anatomy of embryonic chick hearts. This avenue could be extended to image embryonic hearts in more than two channels or with antibody staining, with both being iDSICO+ clearing compatible [51]. Although we primarily focus on regions such as the AAs, dorsal aorta, and outflow tract, our imaging results indicate the light sheet is well suited to capture other fine vessels surrounded by soft tissue, and even various soft tissue types. Imaging additional channels could be used to distinguish soft tissues - such as the cardiac jelly and myocardium - and could be applied to inform mechanical or structural models of early chick hearts and supporting vasculature. Including additional channels could also highlight flow-sensitive genes to complement CFD studies. LSFM has already been used to visualize shear stress-activated signaling pathways, such as Notch signaling in embryonic zebrafish hearts [76]. It would be a natural extension and an advantage of using light sheet fluorescence microscopy for imaging to simultaneously examine gene expression while validating CFD models of subject-specific anatomies. Ultimately, LSFM based imaging methods to inform the computational domain can also be paired with other methods, such as fluorescent RNA *in situ* hybridization [77,78] and/or immunofluorescence, to directly compare mechanosensitive gene/protein expression with the CFD simulations. In this respect, LSFM is an extremely flexible and powerful system generating geometries for computational analysis.

Our CFD model provides a fast and cheap platform to perturb the hemodynamic environment to predict where changes in blood flow might be greatest in the anatomy. Although we apply this pipeline to a cohort of healthy embryonic chicks, the methodology could be expanded to examine how genetic or hemodynamic manipulations can perturb the geometry and the simulated blood flow of embryonic chick hearts at a high visual resolution, particularly in the regions this study identified that experience peak WSS, such as the outflow tract and AAs.

While adult avian and human blood both behave as non-Newtonian fluids, during early stages of development when relatively few red blood cells are circulating, embryonic chick blood is generally considered to be a Newtonian fluid [64]. This study, as well as most CFD studies of these stages and later in embryonic chick development, models blood as a Newtonian fluid [37,79]. However, it may be interesting in the future to examine how altering the modeled embryonic blood to behave as a non-Newtonian fluid would impact the hemodynamics, particularly the spatial and temporal WSS results. Our model currently does not include uncertainty quantification of the results or sensitivity analysis of model parameters. Although CFD studies on the cardiovascular system do not routinely report sensitivity analysis [22], several groups have implemented these analyses for CFD studies of both the mature and developing cardiovascular system [80–83]. In future studies, we would like to examine the sensitivity of input parameters such as the inflow and outflow boundary conditions and the blood viscosity. Additionally, we would also like to conduct an uncertainty quantification such as the generalized polynomial chaos expansion method which has been previously implemented in CFD uncertainty quantification papers to provide a more comprehensive description of probable output values [22,80].

We numerically solved the full incompressible Navier-Stokes equations using the finite element method, which is well-suited for these complicated geometries. The results demonstrating that WSS is elevated within the outflow tract and AAs may suggest that these regions are especially sensitive to heightened and variable WSS and altered blood flow. These findings are consistent with previous ones that report elevated WSS in the AAs [15]. Additionally, the AAs and outflow tract are prone to congenital heart defects [26]. These findings suggest that perturbed hemodynamics may contribute to CHDs that arise in these regions.

Our findings reveal that the AAs are exposed to different wall shear stresses during this stage of development. Radii alone in the AAs was not predictive of peak WSS or blood flow velocity within. Instead, the pair of arches located centrally within the three pairs present, AA III, experienced the highest WSS. The central pair was also the shortest among the pairs present at the stage. These results suggest that geometric features upstream of the arches may influence hemodynamic forces. However, we do recognize that a current limitation of this study is that the model is static, including the OFT. Although the AAs are relativity stiff, *in vivo*, the OFT is relatively dynamic. In future studies we would like to model the OFT to be dynamic, however that is outside the scope of the study.

From these simulations we observed stagnant velocity and WSS spatial patterns across different time points in the cardiac cycle, indicating little convective influence under this flow regime. This is further supported by the little mixing of arch streamlines in the dorsal aorta after traveling through the arches across specimens. To our knowledge, this is also the first report to examine streamlines through each individual arch and trace their distinct behavior down and upstream to other arch streamlines.

Additionally, this cohort-based approach revealed elliptical cross sections and aortic arch diameters on the same order of magnitude as AAs imaged in other cohort-based studies[15,36]. Unlike studies that model blood flow through composite or idealized embryonic heart models, which assume circular cross sections, these individualized studies are the only ones to report elliptical cross sections for AAs at these stages. In modeling hemodynamics in this system, the elliptical shape could be an important feature to consider for future modeling studies that is often ignored.

The combination of using a cohort-based approach and generating a three-dimensional geometric reconstruction allows for an unbiased and comprehensive 3D geometric analysis to be performed. We used the GPA to unbiasedly identify specimens lacking anatomical features such as AAs. Our findings through the visualization techniques of the GPA results indicate different geometric features such as the outflow tract and AA II are more likely to vary compared to other geometric features. The GPA also provides an unbiased method to identify outliers for further downstream analysis. In addition to conducting a sensitivity analysis on physiological model parameters such as inflow and outflow boundary conditions, we would extend the analysis to examine the impact of variations of model geometry on output. The GMM analysis could be used as an unbiased but directed method to identify which morphological features vary the most, and these features could be included in the sensitivity analysis.

It is well established that hemodynamics plays a critical role in shaping the course of a developing heart in humans as well as other organisms with four-chambered hearts. Unfortunately, technical limitations preventing direct measurements of blood flow patterns within animal model hearts have stagnated the progress to elucidating the role of hemodynamics in cardiovascular development. This flexible and customizable pipeline overcomes this challenge by using high-resolution imaging to inform CFD modeling simulating blood flow within reconstructed embryonic chick hearts. CFD modeling of this system is not novel in-and-of itself, however, we have attempted to develop methods that are fast and easily customizable for a broad variety of hemodynamic studies across specimen-specific morphologies. This pipeline can easily be modified and extended to explore various aspects of cardiovascular development in embryonic chicks and other model organisms and to conduct robust sensitivity analysis on flow parameters and model geometry across individual morphologies.

In conclusion, this study demonstrates the utility of pairing advanced imaging techniques with computational models and GMMs by using light sheet fluorescence microscopy imaging to generate subject-specific CFD analysis in embryonic chick studies. Our findings contribute to the growing body of literature on CFD models of embryonic chick AAs by providing a fast and cheap imaging method that can easily be used to generate cell-accurate geometries. Ultimately the CFD and morphometric analysis can be extended to compare hemodynamic and morphological features in several ways, including overlaying gene expression patterns with flow sensitive regions. Methods for comparing flow dynamics within individual morphologies will furthermore allow for translations to human fetal heart analysis.

## Methods

### Chick embryo preparation

Fertilized White Leghorn (*Gallus domesticus*) eggs were incubated in a humidified incubator at 38°C to the desired developmental stages of approximately Hamburger-Hamilton (HH) 16–17 [84], which was around 2.5 days [85]. Development stages were confirmed using morphological features [86]. Each eggshell was windowed, and the shell membrane removed to expose the embryo. 1–2 ml of phosphate-buffered saline was added to each egg with parafilm placed over the window to prevent the embryos from dehydrating. Following the described micro-injections, the embryos were removed from the eggs and isolated in Hanks Buffered Saline Solution containing 20 mM 2,3 butanedione monoxime for ten minutes, which arrested cardiac motion. Embryos were then transferred into 6-well plates containing ice cold 4% PFA. The embryos were shaken in PFA overnight at 4°C prior to clearing. Each specimen was cleared using the iDISCO+ protocol with the alternative pretreatment and stored in glass vials in dibenzyl ether at 4°C prior to imaging. The light sheet fluorescence microscope required placing the sample in a dibenzyl ether-filled chamber [87]. The specimen was secured within a cradle using a screw, with the cradle then fitted in the chamber. Typically if samples are too small to be positioned using the screw, such as the embryos in this study, they are alternatively embedded in agarose [88]. However, agarose embedding has the potential to distort cardiac and vascular geometry. Therefore, we constructed hollow agarose boxes with removable lids to house singular embryos during clearing and imaging. Each embryo was placed in an agarose box, secured with agarose lids molded from custom 3D-printed molds designed in Fusion360 (Autodesk) [89].

### Imaging and post-processing

Chick embryos were imaged at the UNC Microscopy Services Laboratory using the LaVision BioTec Ultramicroscope II. The Imaris (Bitplane, Oxford Instruments) File Converter software was used to convert light sheet fluorescence microscopy-generated TIFF files to Imaris files.

### Computational model

Grid spacing was approximately 0.009 mm with each embryo containing at least 100,000 second-order tetrahedral elements. Blood flow was assumed to be laminar. Four cardiac cycles were completed for each simulation with a time step size of 0.01 s.

## Post processing and analysis

Simulations and resulting hemodynamics were visualized in Paraview (Kitware) [75]. Aortic arch centerline extraction was performed using the vascular modeling toolkit module VMTKSlicer extension (https://github.com/vmtk/SlicerExtension-VMTK) [63,90,91] in the open-source software 3D Slicer (https://www.slicer.org/) [59] to determine the length, tortuosity, curvature, average radii, and other geometric properties of each aortic arch. Radii were determined by calculating the maximally inscribed sphere along points in the arches. Geometric visualization and analyses were performed using RStudio (https://posit.co/) in R (https://www.r-project.org/).

## Supporting information

**S1 Fig. Basic geometric features can be extracted from segmentations derived from microscopy images.** (A) Following manual segmentation from imaging, the region of interest was selected which included the outflow tract, AAs, and dorsal aorta. Next the segmentation was smoothed, solidified and discretized into a tetrahedral mesh. (B) Aortic arch cross sections are elliptical in shape at this stage. (C) AA III contains the largest average diameter, while AA IV contains the smallest minimum and average diameter. (D-G) Arches were compared laterally (left/right sides) and pairwise (AA II/AA III/ AA IV) using ANOVA followed by Tukey analysis (where * indicates $P < 0.05$, ** indicates $P < 0.01$, *** indicates $P < 0.001$). In general, arches varied by pair, but not laterally (left vs. right). Diameters were calculated by measuring the diameter of the maximally inscribed sphere (MIS) at a given point in the arch. (D) The central arch, AA III was significantly shorter than either of the other arches flanking it. (E) AA IV had the smallest minimum diameter among the three pairs and was significantly smaller than the minimum diameter of AA III. (F) AA IV contained the smallest average diameter and had a significantly smaller average diameter compared to AA III. (G) Aortic arch volumes did not significantly differ by side or pair.
(TIF)

**S2 Fig. Streamline tracing in case 2 indicates streamlines follow a similar pattern to general cases.** In case 2, blood flow is simulated to flow out the cranial arches in addition to the dorsal aorta. Case 2 streamlines are shown next to a general case for comparison. (A) Lateral and cranial views of case 2 compared to the general case are shown. Streamlines from R II, R III, L II, and L III can be observed exiting through the cranial arteries. (B) The cross sections of the outflow tract and dorsal aorta are shown. Streamlines for RII and L II do not exit through the dorsal aorta in case 2, however the remaining streamlines maintain the same spatial orientation compared to the general case in the absence of L II and R II streamlines.
(TIF)

**S3 Fig. Mesh convergence study.** (A) We constructed increasingly fine meshes of a template embryo ranging from 43,587 tetrahedral elements to 301,736 elements. (B) We examined simulation results under these different levels of mesh refinement by sampling velocity magnitude across a line drawn across the lumen shown above. Cross sections of the lumen with the simulations under the different mesh refinements are also shown with the lines that were used to sample the velocity.
(TIF)

## Acknowledgments

Light sheet fluorescence microscopy was performed at the Microscopy Services Laboratory, Department of Pathology and Laboratory Medicine. Imaris segmentation was performed using the Imaris software at the UNC Hooker Imaging Core Facility. All figures were created in BioRender. Giesbrecht, K. (2025) https://BioRender.com/k26d165. Thank you to Kathryn Scherrer for proof-reading and editing the manuscript.

## Author contributions

**Conceptualization:** Kirsten Giesbrecht, Michael Bressan, Boyce E. Griffith, Simone Rossi.

**Data curation:** Shourya Mukherjee, Sophie Liu.

**Formal analysis:** Kirsten Giesbrecht.

**Investigation:** Kirsten Giesbrecht, Simone Rossi.

**Methodology:** Boyce E. Griffith, Kirsten Giesbrecht, Michael Bressan, Simone Rossi.

**Project administration:** Boyce E. Griffith.

**Resources:** Michael Bressan, Boyce E. Griffith.

**Software:** Kirsten Giesbrecht, Simone Rossi.

**Supervision:** Michael Bressan, Boyce E. Griffith.

**Visualization:** Kirsten Giesbrecht.

**Writing – original draft:** Kirsten Giesbrecht, Michael Bressan.

**Writing – review & editing:** Michael Bressan, Boyce E. Griffith.

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
