## [Decision Letter · Decision Letter 0]

17 Jan 2025

PONE-D-24-51917An anatomically informed computational fluid dynamics modeling approach for quantifying hemodynamics in the developing heartPLOS ONE

Dear Dr. Bressan,

Thank you for submitting your manuscript to PLOS ONE. After careful consideration, we feel that it has merit but does not fully meet PLOS ONE’s publication criteria as it currently stands. Therefore, we invite you to submit a revised version of the manuscript that addresses the points raised during the review process. Please submit your revised manuscript by Mar 03 2025 11:59PM. If you will need more time than this to complete your revisions, please reply to this message or contact the journal office at plosone@plos.org . Please include the following items when submitting your revised manuscript:

We look forward to receiving your revised manuscript.

Kind regards,

Adélia Sequeira, Ph.D

Academic Editor

PLOS ONE

**Journal Requirements:**

This work is supported by the American Heart Association Predoctoral fellowship 899419 to K Giesbrecht. 

Light sheet fluorescence microscopy was performed at the Microscopy Services Laboratory, Department of Pathology and Laboratory Medicine. Research reported in this publication was supported in part by the North Carolina Biotech Center Institutional Support Grant 2016-IDG-1016. Imaris segmentation was performed using the Imaris software at the UNC Hooker Imaging Core Facility. The UNC Hooker Imaging Core Facility and the Microscopy Services Laboratory, Department of Pathology and Laboratory Medicine are both supported in part by P30 CA016086 Cancer Center Core Support Grant to the UNC Lineberger Comprehensive Cancer Center. All figures were created in BioRender. Giesbrecht, K. (2025) https://BioRender.com/k26d165. Thank you to Kathryn Scherrer for proof-reading and editing the manuscript. 

This work is supported by the American Heart Association Predoctoral fellowship 899419 to K Giesbrecht. 

**Additional Editor Comments:**

The manuscript presents interesting results on a very relevant topic and deserves to be published. However, it undergoes several weaknesses outlined by both reviewers. One of the reviewers addressed more critical remarks than the other one, but both pointed out important issues that should be considered by the authors.

In my opinion, the authors need to rework the present manuscript by reformulating some statements in order to incorporate in the text most of the reviewer’s observations. References to previous works should be included and, of course, more limitations to the present findings should also be identified in the manuscript.

Reviewers' comments:

Reviewer's Responses to Questions

**Comments to the Author**

1. Is the manuscript technically sound, and do the data support the conclusions?

Reviewer #1: Yes

Reviewer #2: No

2. Has the statistical analysis been performed appropriately and rigorously? 

Reviewer #1: N/A

Reviewer #2: N/A

3. Have the authors made all data underlying the findings in their manuscript fully available?

Reviewer #1: Yes

Reviewer #2: Yes

4. Is the manuscript presented in an intelligible fashion and written in standard English?

Reviewer #1: Yes

Reviewer #2: Yes

5. Review Comments to the Author

**Reviewer #1: ** This paper deals with the challenge of understanding congenital heart defects (CHDs). Less than a third of CHDs have known genetic or environmental causes. Hemodynamic forces, such as wall shear stress, influence heart development but are difficult to measure in vivo. To address this, the authors developed a computational fluid dynamics (CFD) modeling pipeline. This approach numerically simulates blood flow and computes hemodynamic forces in 3D reconstructions of embryonic chick hearts. The method integrates light sheet microscopy and geometric morphometric analysis for anatomical variability. The authors claim that the approach offers a fast, robust tool to study the origins of CHDs.

The paper is technically sound, meaningful, and relevant, presenting new ideas and concepts. Overall, it merits publication, provided the following questions are addressed. 1.The computational pipeline is applied to chick embryos. How can the findings and methodology be translated to humans? This includes not only the procedural aspects but also the implications of the findings, which remain unclear. 2.Regarding the numerical simulations, it is stated that a laminar flow model is considered. Does this imply that no turbulence model is employed? However, would a turbulence model be truly necessary given the nature of blood flow and the resolution of the computational mesh? 3.Unsteady flow is considered in a fixed computational domain. Is it correct? 4.The inflow and outflow boundary conditions are briefly mentioned, but their implementation lacks clarity. Given that the flow is highly sensitive to these boundary conditions, the numerical results—particularly the mean velocity and wall shear stress—can be significantly influenced by their choice. How robust are the results and, consequently, the study’s findings under these assumptions? 5.Geometrical imprecision in the geometrical model can significantly impact the results of numerical simulations. How do such inaccuracies affect the robustness and reliability of the findings? Has this been quantified or accounted for in the analysis?

**Reviewer #2:**  This paper represents an attempt at quantifying morphology and blood flow in a developing chick embryo extraembryonic vessels. The authors developed a procedure that uses light sheet microscopy to image the chick embryo vessels at early stages of development (HH16-HH17), when the heart is still tubular and connects to the aortic sac from where aortic arches branch that then end in the dorsal aorta. The anatomy of the arches has been described before, and the authors use their developed procedures to show variations in vessel morphology among embryos using morphometric methods. They then use the reconstructed 3D vessel geometry to run CFD simulations to determine flow characteristics.

Unfortunately, some of the procedures employed by the authors are not properly implemented. This, together with lack of proper knowledge of previous works leads to over-exagerated claims for the results achieved, as well as the innovation of the work.

Similar work - using slightly different tools - has been implemented by several groups, including the Butcher group at Cornell, and Pekkan and Keller groups. Light sheet microscopy has also been implemented in the living zebra fish embryo heart - and CFD models (with moving walls - more advanced than the ones by the author) have been developed by Marsden at Stanford and Yap in the UK. 4D reconstructions of the heart have also been achieved using OCT imaging by several groups, including groups at Baylor and Case Western, and wall shear stresses have been estimated from measured flow and through CFD of the moving heart. All these works are grossly minimized or neglected by the authors, who point to "lack of protocols" - and who claim their methods are superior, as nobody could before calculate flow or wall shear stress. In fact their Table 1 omits several works.

CFD models employed by the authors are not validated, and meshes employed are not rigorously tested through a convergence study. Moreover, the geometry employed and boundary conditions do not seem appropriate. CFD boundary conditions are not shown - and comparisons with actual velocity measurements are not used to validate the model results.

The authors start talking about the heart and flow in the heart - but end up modeling only the heart outflow tract and the extracardiac aortic arches and dorsal aorta. It would be more appropriate to neglect the heart outflow tract portion, as it is wrongly captured from fixed embryos (see below). In the authors' models, it would make more sense to start with the aortic sac, then model the conection to the arches and then the aorta. All these are relatively 'fix' portions of vasculature in the embryos.

The outflow tract is a very dynamic cardiac structure: it opens and closes during the cardiac cycle as blood flows through it. But instead of modeling this motion, the authors chose a static (fix) geometry. While this might be a fair approximation (and certainly employed before from CT images) - the procedures employed by the authors to fix the embryos do not seem to arrest the heart properly. The authors mention "the degree of stenosis in the outflow tract" that is apparently different among samples. This likely reflects the fact that the outflow tract of different embryos is arrested at different points in the cardiac cycle (e.g. open outflow tract, semi-closed outflow tract) given the impression that sometimes the outflow tract has "stenosis". There is no "stenosis" in a normal outflow tract - what the authors observed are the outflow tract cushions that are more or less obvious depending on how close or open the outflow tract walls are. The authors should be aware of anatomy and motion of the outflow tract and its implications before attempting to model it. In their models, it would be simpler to neglect the outflow tract.

Authors also mention that vessels have an elliptical cross-sections - how do they know this is not an artifact from the embryo fixation, and/or imaging?

Later in the paper, the authors are then "surprised" to find that regions of high flow velocities also corresponds to regions of high wall shear stress, and that inertia effects are negligible. This is simply due to the low flow Reynolds number. This is not surprising given the small size of the embryo and its heart adn vasculature.

Overall, the more interesting contribution of the paper is in the morphological evaluations of the arches and dorsal aorta anatomy. The CFD models are interesting, but their implementation is not appropriate and simulations have not been validated, nor rigorously tested.

6. PLOS authors have the option to publish the peer review history of their article (what does this mean? ). If published, this will include your full peer review and any attached files.

**Do you want your identity to be public for this peer review?** For information about this choice, including consent withdrawal, please see our Privacy Policy .

Reviewer #1: No

Reviewer #2: No

---

## [Author Response · Author response to Decision Letter 1]

6 Mar 2025

As requested in the Decision Letter sent on Friday January 17th, we have created a point-by-point response to the reviewer comments and uploaded that response to the revised submission package.

---

## [Editor Report · Decision Letter 1]

18 Mar 2025

An anatomically informed computational fluid dynamics modeling approach for quantifying hemodynamics in the developing heart

PONE-D-24-51917R1

Dear Dr. Bressan,

We’re pleased to inform you that your manuscript has been judged scientifically suitable for publication and will be formally accepted for publication once it meets all outstanding technical requirements.

Kind regards,

Adélia Sequeira, Ph.D

Academic Editor

PLOS ONE

Additional Editor Comments (optional):

The authors have adequately responded to the comments raised by the reviewers.

The revised version of the manuscript has been significantly improved. 
---

## [Editor Report · Acceptance letter]

PONE-D-24-51917R1

PLOS ONE

Dear Dr. Bressan,

I'm pleased to inform you that your manuscript has been deemed suitable for publication in PLOS ONE. Congratulations! Your manuscript is now being handed over to our production team.

Kind regards,

on behalf of

Dr. Adélia Sequeira

Academic Editor

PLOS ONE